Mapping the global distribution of invasive pest Drosophila suzukii and parasitoid Leptopilina japonica: implications for biological control

Nair Rahul R. rahulraveendran@ku.edu rahulravi777@gmail.com
Peterson A. Townsend
Biodiversity Institute, University of Kansas , Lawrence , KS , United States of America
Silva Daniel
Electronic publication date: 2023 Apr 24
Publication date: 2023
Volume: 11
Electronic Location ID: e15222
Received 2022 Dec 6; Accepted 2023 Mar 22
Copyright: ©2023 Nair and Peterson
Copyright year: 2023
Copyright holder: Nair and Peterson
License: This is an open access article distributed under the terms of the Creative Commons Attribution License, which permits unrestricted use, distribution, reproduction and adaptation in any medium and for any purpose provided that it is properly attributed. For attribution, the original author(s), title, publication source (PeerJ) and either DOI or URL of the article must be cited.
License URL: https://creativecommons.org/licenses/by/4.0/

Keywords: Drosophila suzukii, Leptopilina japonica, Pest, Parasitoid, Invasion, Biological control, Ecological niche modeling

Funding: Mr. Ajayya Kumar, COO, Emircom, Abu Dhabi Rahul R. Nair was financially supported by Mr. Ajayya Kumar (COO, Emircom, Abu Dhabi). The funders had no role in study design, data collection and analysis, decision to publish, or preparation of the manuscript.

==============================
Insect pest invasions cause significant damage to crop yields, and the resultant economic losses are truly alarming. Climate change and trade liberalization have opened new ways of pest invasions. Given the consumer preference towards organic agricultural products and environment-friendly nature of natural pest control strategies, biological control is considered to be one of the potential options for managing invasive insect pests. Drosophila suzukii (Drosophilidae) is an extremely damaging fruit pest, demanding development of effective and sustainable biological control strategies. In this study, we assessed the potential of the parasitoid Leptopilina japonica (Figitidae) as a biocontrol agent for D. suzukii using ecological niche modeling approaches. We developed global-scale models for both pest and parasitoid to identify four components necessary to derive a niche based, target oriented prioritization approach to plan biological control programs for D. suzukii: (i) potential distribution of pest D. suzukii, (ii) potential distribution of parasitoid L. japonica, (iii) the degree of overlap in potential distributions of pest and parasitoid, and (iv) biocontrol potential of this system for each country. Overlapping suitable areas of pest and parasitoid were identified at two different thresholds and at the most desirable threshold (E = 5%), potential for L. japonica mediated biocontrol management existed in 125 countries covering 1.87 × 107 km2, and at the maximum permitted threshold (E = 10%), land coverage was reduced to 1.44 × 107 km2 in 121 countries. Fly pest distributional information as a predictor variable was not found to be improving parasitoid model performance, and globally, only in half of the countries, >50% biocontrol coverage was estimated. We therefore suggest that niche specificities of both pest and parasitoid must be included in site-specific release planning of L. japonica for effective biocontrol management aimed at D. suzukii. This study can be extended to design cost-effective pre-assessment strategies for implementing any biological control management program.

Introduction

Over recent decades, the world has witnessed significant increases in agricultural production, but increases in crop yields have often been reduced by diverse insect pests (Vreysen et al., 2007; Savary et al., 2019). Assessment of all of the components of agricultural productivity and food security must include consideration of insect pests, as they are an integral part of anthropogenic crop ecosystems (Food and Agriculture Organization, 2013; Savary et al., 2019). Global warming and economic globalization accelerate development of new routes of pest invasion (Girod et al., 2018), presenting new challenges. As pests pose serious threats in the functioning of global food systems (Savary et al., 2017), various strategies have been developed for insect pest management, each with its own advantages and disadvantages (Dara, 2021). Improvement in the management of invasive pest populations includes consideration of sustainable and eco-friendly approaches, with the goal of achieving long-term benefits (Bernaola & Holt, 2021).

A broad (fruit) host range (Lee et al., 2011; Bellamy, Sisterson & Walse, 2013), combined with an ability to infest ripening soft fruits (Gabarra et al., 2015), has made Drosophila suzukii (Matsumura) (Diptera: Drosophilidae) an economically damaging, globally invasive fruit pest of serious concern (Walsh et al., 2011). Preference for not-quite-ripe or just-ripe fruits over damaged or decaying fruits (Mitsui, Takahashi & Kimura, 2006), and the presence of a sclerotized ovipositor of females (Kienzle & Rohlfs, 2021) with serrations to pierce undamaged fruit epicarps for laying eggs, are two notable traits (Walsh et al., 2011) that contribute significantly to economic threats imposed by D. suzukii. Bacterial and fungal pathogens can cause secondary infections in fruits after infestation by D. suzukii, augmenting economic losses (Molina, Harrison & Brewer, 1974; Louis et al., 1996; Walsh et al., 2011).

Drosophila suzukii is native to eastern and southeastern Asia (Bolda, Goodhue & Zalom, 2010); it was initially detected in Japan in 1916 (Kanzawa, 1935) and described as a distinct species in 1931 (Hauser, 2011). In 2008, D. suzukii was identified as an invasive species for the first time with populations in both North America (Hauser, 2011) and Europe (Calabria et al., 2012). Its host range covers 13 angiosperm families (Cloonan et al., 2018), and its invaded geographic range has now extended to South America (Deprá et al., 2014; Andreazza et al., 2017) and Africa (Kwadha et al., 2021). As D. suzukii larvae feed inside of fruits (Fanning, Grieshop & Isaacs, 2018), and the fruit export trade strictly follows zero-tolerance towards infestations (Tait et al., 2021), much high-value fruit is rendered unsellable every year. Economic impact assessments in the United States (Bolda, Goodhue & Zalom, 2010; Walsh et al., 2011; Goodhue et al., 2011; Farnsworth et al., 2017; DiGiacomo et al., 2019; Yeh et al., 2020), Europe (Knapp, Mazzi & Finger, 2021), and South America (Benito, Lopes-da Silva & Santos, 2016), have indicated losses on the order of US$550M per year.

Various preventive and post-infestation control measures (Lee et al., 2011; Landolt, Adams & Rogg, 2012; Haye et al., 2016; Schetelig et al., 2018; Shawer et al., 2018; Tait et al., 2021) have been developed so far, but none with complete efficacy (Kehrli et al., 2017; Knapp, Mazzi & Finger, 2019). Management strategies for D. suzukii can be classified broadly into four categories: (1) chemical control (Beers et al., 2011; Van Timmeren & Isaacs, 2013; Shawer et al., 2018; Shawer, 2020), (2) microclimate manipulation (Lee et al., 2016; Rendon et al., 2020), (3) RNA interference biopesticides (Murphy et al., 2016), and (4) biological control (Chabert et al., 2012; Daane et al., 2016; Mazzetto et al., 2016; Knoll et al., 2017; Daane et al., 2021). Extensive use of chemical methods to control D. suzukii infestations can lead to increased pest resistance, and concerns regarding food and environmental safety (Santoiemma et al., 2019). Microclimate manipulation approaches to control D. suzukii are more likely to perform well in hot and dry regions (Schöneberg et al., 2022), as D. suzukii is sensitive to high temperatures and low humidity (Rendon et al., 2020). RNA interference methods involve higher development costs and involve much labor (Bramlett, Plaetinck & Maienfisch, 2020). Finally, biological control involves release of enemies of D. suzukii from the region of its origin (Asia) in invaded areas, as a means to reduce its population growth (Girod et al., 2018). This method is recommended (Cock et al., 2010; Van Lenteren, 2012) in view of improved food safety, environment-friendly characteristics, economic feasibility, and long-term control solutions that are established (Kruitwagen, Beukeboom & Wertheim, 2018).

Parasitoid wasps of the genera Asobara (Braconidae), Ganaspis (Figitidae), and Leptopilina (Figitidae) have been studied extensively as biological control agents with potential to suppress growth of D. suzukii populations (Kacsoh & Schlenke, 2012; Rossi Stacconi et al., 2015; Daane et al., 2016; Giorgini et al., 2019; Wang et al., 2019; Biondi, Wang & Daane, 2021). In particular, the species A. japonica, G. brasiliensis, and L. japonica are potential biocontrol agents (Wang et al., 2019). However, some researchers do not recommend A. japonica for biological control programs aimed at D. suzukii (Daane et al., 2016; Girod et al., 2018; Abram et al., 2020), owing to its broad host range (Ideo et al., 2008; Furihata et al., 2016). Indeed, given its host specificity, G. brasiliensis has been proposed as a candidate for biological control of D. suzukii (Wang et al., 2020); yet, in a scenario when these three wasps coexist, L. japonica is unique in being able to outcompete the other two species thanks to its relatively faster egg-hatching potential (Wang et al., 2019). Relatively high host specificity (Wang et al., 2020), demonstrated competence in multi-parasite systems (Wang et al., 2019), and recent range expansions into areas invaded by D. suzukii in Europe (Puppato et al., 2020) and North America (Abram et al., 2020; Abram et al., 2022), make L. japonica an intriguing candidate biocontrol agent for D. suzukii that can be tested for overall effectiveness.

Ecological niche modeling (ENM) has been used extensively to identify potential distributions of species for a variety of purposes (Raxworthy et al., 2007; Escobar, 2020; Kolanowska & Jakubska-Busse, 2020; Wan et al., 2020; Valencia-Rodríguez et al., 2021; Agboka et al., 2022; Demján et al., 2022; Outammassine, Zouhair & Loqman, 2022). In pest-parasitoid systems, identifying and comparing relative habitat suitability of pest and parasitoid can help to guide effective biological control programs (Pérez-de la O et al., 2020; Tepa-Yotto et al., 2021a; Tepa-Yotto et al., 2021b). The utility of ENM in applications to biological control of pests can be attributed to two factors: alien parasitoid species must survive and reproduce in the geographic regions where they are released (Mills, 2018; Schulz, Lucardi & Marsico, 2019), and unfavorable abiotic factors can reduce the long-term efficacy of biological control measures (Olfert et al., 2016). Modeling climatic preferences of deliberately introduced parasitoid species can also provide insights into possible range expansions, an important aspect to be tested in improving effectiveness of classical biological control programs (Pérez-de la O et al., 2020).

In this study, we used ENM approaches to explore, discuss, and highlight five aspects of a biological control strategy for D. suzukii that can directly benefit producers, extension agents, and policy makers. (1) We estimated the potential distribution of the invasive pest D. suzukii, and (2) that of the parasitoid L. japonica. (3) We assessed the degree of overlap in the potential distributions of D. suzukii and L. japonica, and (4) estimated the biocontrol potential of this system for each country. Finally, (5) we assessed parasitoid model performance to see if incorporating distributional information for the pest improves model performance for the parasitoid.

Methods

Occurrence data

Occurrence records of D. suzukii were downloaded from five online biodiversity data portals: Global Biodiversity Information Facility (GBIF; http://www.gbif.org, accessed on 2 August, 2022; DOI: https://doi.org/10.15468/dl.hxg8z2), Biodiversity Information Serving Our Nation (BISON; http://www.gbif.us, accessed on 2 August, 2022), Berkeley Ecoinformatics Engine (Ecoengine; ecoengine.berkeley.edu, accessed on 2 August, 2022), iNaturalist (http://www.inaturalist.org, accessed on 2 August, 2022), and Integrated Digitized Biocollections (iDigBio; http://www.idigbio.org, accessed on 2 August, 2022) using Spocc version 1.2.0 R package (Chamberlain, Ram & Hart, 2021); occurrence data were also drawn from the Centre for Agriculture and Bioscience International (CABI; http://www.cabi.org, accessed on 3 August, 2022), and published literature (see File S1 for details). This initial harvest of occurrence data yielded an initial total of 2369 records.

A five-step data cleaning process was adopted: (1) removal of records with no date of observation, (2) removal of incomplete coordinates (i.e., lacking valid latitude and longitude), (3) removal of unlikely coordinates (e.g., 0.00°N, 0.00°E), (4) removal of duplicated coordinates, and (5) removal of coordinates with fewer than two decimal places. Data cleaning was performed using scrubr version 0.1.1 R package (Chamberlain, 2016). The cleaned dataset (1385 records) was overlaid on climatic raster layers (5′ or ∼10 km spatial resolution, see below) to remove points falling outside the raster boundaries. The resulting occurrence dataset (1377 records) was subjected to visual inspection to detect clusters of points (often related to points of access or concentrations of people), and eliminate disproportionate data density at random, maintaining a minimum distance of ≥ 30km among points, to avoid model overfitting (Raghavan et al., 2019). The final dataset of 314 points (Fig. 1; File S1) showed no excessive clustering of occurrences across the known distribution of D. suzukii. Spatial filtering was performed using spThin R package (Aiello-Lammens et al., 2015).

Figure 1 Distributional information.

Representation of the known distribution of the pest Drosophila suzukii, and parasitoid Leptopilina japonica based on occurrence databases and published literature.

Occurrence records of L. japonica were sourced from published literature (Abram et al., 2022; Wang et al., 2022; Abram et al., 2020; Puppato et al., 2020; Giorgini et al., 2019; Girod, 2018; Novković et al., 2011), as online data portals held few or no records. A distance filter of 12 km was applied to the occurrences extracted, and the final dataset comprised 31 points (Fig. 1; File S2). Leptopilina japonica has two subspecies: L. japonica japonica and L. japonica formosana, occurring in Japan and Taiwan respectively (Novković et al., 2011); both have the ability to parasitize D. suzukii (Kimura & Novković, 2015). Our final dataset included mostly the nominate subspecies, and only a single occurrence record of L. j. formosana (Novković et al., 2011).

Environmental data

Bioclimatic raster layers at 5′ spatial resolution (∼10 km at the Equator) were downloaded from WorldClim 2.1 for present conditions (1975–2000; Fick & Hijmans, 2017). Variables combining temperature and precipitation measurements (i.e., mean temperature of wettest quarter, mean temperature of driest quarter, precipitation of warmest quarter, and precipitation of coldest quarter) were excluded (Escobar et al., 2014) owing to discontinuous patterns of those variables in many areas (Booth, 2022).

To define the set of limits and conditions for ENM, identification of areas accessible to species over relevant time periods (Soberón & Peterson, 2005; Peterson & Soberón, 2012) is essential to development of robust models (Barve et al., 2011). The development of a hypothesis of accessible area M is crucial for rigorous characterization of niche characteristics of species (Barve et al., 2011; Machado-Stredel, Cobos & Peterson, 2021). Considering the near-global distribution of D. suzukii and L. japonica, the entire world (excluding Antarctica) was defined as the accessible area for the two species. The 15 climatic data layers were clipped to the extent of this area. Multi-collinearity and dimensionality among the clipped bioclimatic layers were minimized using principal components analysis, in effect transforming correlated climatic variables into fewer, uncorrelated principal components (PCs), and these multivariate environmental variables were used as the independent variables in ENM.

The advantage of principal component analysis over other methods of multi-collinearity reduction is that a significant proportion of all original information related to variables can be retained in the form of independent components (Tabachnick & Fidell, 2007; Cruz-Cárdenas et al., 2014), summarizing all environmental variation across a particular geographic region (Júnior & Nóbrega, 2018). Each PC is a linear combination of all of the 15 original climatic variables: the first PC summarizes the major axis of the multivariate space, explaining a large proportion of the total variance in the original data; the second PC explains a maximum of the remaining variance, which is independent of the first axis; and so on (Cruz-Cárdenas et al., 2014). Principal components analysis of raster variables was done using the kuenm_rpca function of the kuenm R package (Cobos et al., 2019). Contributions of each of the original bioclimatic variables to the PCs (Sillero et al., 2021) and average contribution of each of the PCs to the final models of pest and parasitoid were estimated (Quiner & Nakazawa, 2017), to have insight into important variables driving niches and distributions of pest and parasitoid. We applied an arbitrary threshold of absolute value of factor loadings to assess the relative importance of variables to each of the PCs (Rotenberry, Preston & Knick, 2006; Barrows et al., 2008); variables with factor loadings ≥0.35 were explored as potentially important (Bogosian III et al., 2012). Loading values of the variables represent the extent to which those variables are correlated with particular PCs (Júnior & Nóbrega, 2018). In the case of PCs with mixed positive and negative loadings, variables with positive loadings >0.35 contribute the same amount of information as that of variables with negative loading <−0.35, as, in both cases, the absolute value of loadings exceeds our arbitrary threshold of 0.35. The signs of the loadings indicate the nature of correlation of variables with the PCs.

Ecological niche modeling

In separate ENM analyses, occurrences of each species (pest and parasitoid) were partitioned randomly into training and testing data in two different proportions: 70:30 for D. suzukii, and 50:50 for L. japonica. Considering the small number of records, data-splitting ratio was reduced to 0.5 for L. japonica to maintain a balance between predictive accuracy and performance estimation of models as very low sample size for testing can cause errors in estimating predictive accuracy (Peterson, Ball & Cohoon, 2002). Modeling experiments were performed using six combinations of three feature classes (l-linear, q-quadratic, p-product; l, q, lq, qp, lp, and lqp; product response types were not used in isolation owing to occasional problems that result), 10 regularization multipliers (0.1, 0.3, 0.6, 0.9, 1, 2, 3, 4, 5, 6), and nine sets of principal components summarizing climate data. The first 10 PCs accounted for >99% of the total variation: set 1 (PCs 1 and 2), set 2 (PCs 1-3), etc., up to set 9 (PCs 1-10). Best models were selected by applying three criteria sequentially (Cobos et al., 2019): (1) choosing statistically significant models using partial ROC tests, (2) filtering statistically significant models to those with <5% omission error (E), and (3) ranking all remaining models based on Akaike information criterion (AICc) values; the subset of significant, low-omission models within 2 AICc units of the minimum were selected as the best models (Warren & Seifert, 2011).

Mean AUC ratios of bootstrap replicate models were calculated using the partial ROC approach, which remedies some of the known problems with traditional receiver operating curve (ROC) analysis (Peterson, Papeş & Soberón, 2008). In this method, the importance of negative (absence) information is reduced, as such information is generally unavailable (Peterson, Papeş & Soberón, 2008). Crucially, the interpretation of the area under the curve is limited to relevant portions of the curve, that is those parts meeting user-defined low omission thresholds, in this study E = 5%. Then, as one is generally not evaluating the curves over the entire space, to assess statistical significance, AUC ratios are defined as the ratio of AUC of the partial ROC curve to the area under the random expectation line over the same restricted part of the space. AUC ratio values range from 0 to 2; a value of 1 indicates random performance (Peterson, Papeş & Soberón, 2008; Peterson, 2012). Model fitting was replicated 10 times using bootstrapped subsamples of the available occurrence data; variation among replicates was then used to assess whether the AUC ratio exceeds 1 significantly, and the median of the median suitability outputs across all replicates was used to interpret results for each species.

To assess the potential role of fly distributional information in improving the performance of the parasitoid model, the final D. suzukii model output was added to each multivariate environmental variable set. We then re-calibrated the L. japonica model using the same set of feature class types and regularization multiplier values to develop a two-species model for the wasp (see Ashraf, Chaudhry & Peterson, 2021). We compared models with and without the fly distributional information using the same 3 criteria described above. Occurrence data partitioning exercises were done using caTools R package (Tuszynski, 2021). All modeling experiments were performed using maximum entropy approaches (Maxent) (Phillips, Anderson & Schapire, 2006), as implemented in the kuenm R package (Cobos et al., 2019).

To represent suitable and unsuitable regions for the pest and the parasitoid, Maxent models in the form of continuous logistic outputs were transformed into binary presence-absence models by applying two different least-training presence thresholds (i.e., allowable omission E = 5% and E = 10%). These two thresholds were chosen as indices of most desirable (E = 5%) and maximum permitted (E = 10%) omission rates to represent relative habitat suitability, and also to avoid overinterpretation of predictions (Ashraf, Chaudhry & Peterson, 2021). These thresholds were applied using QGIS Tisler desktop version 3.24.3 (QGIS Geographic Information System, 2022).

Similarity between niche estimates for pest and parasitoid was quantified using Schoener’s D index based on two methods: an ENM-based method which compares niches in geographic space (Warren, Glor & Turelli, 2008) and a parallel, ordination-based method (PCA-env) that compares niches in environmental space using similar tests (Broennimann et al., 2012). Given that the area M was same for both pest and parasitoid, a symmetric background similarity test (Warren, Glor & Turelli, 2008; Warren et al., 2021) was used to implement the ENM-based method. The observed D value of the two empirical models (pest and parasitoid) was compared with a null distribution of D values generated by comparing the expected overlap of the niche estimates of 100 replicates of pest and parasitoid models, developed by drawing random occurrences 100 times from the background of both species, retaining the original sample sizes (Warren, Glor & Turelli, 2008; Warren et al., 2021). In environmental space, the PCA-env method (Broennimann et al., 2012) was used to summarize climatic variability across the M area of both species. This method tests whether the niche occupied by pest is similar to that occupied by parasitoid. Occurrence densities of pest and parasitoid were shifted randomly 100 times in the background, and niche overlap was calculated in each iteration to create a null distribution of D values. In both cases, we used a one-tailed test focusing on rejecting a null hypothesis of niche similarity (Peterson, 2011; Tocchio et al., 2015; Qiao, Escobar & Peterson, 2017), and ignoring the upper tail of the distribution that would be significant niche similarity, which is of unknown biological meaning. Non-rejection of the null hypothesis of niche similarity arises when the empirical D value falls within the upper 95% of the null distribution of D values (P > 0.05) (Tocchio et al., 2015), indicating that niches of the two species are not demonstrably distinguishable. We used the ENMTools 1.0 (Warren et al., 2021) and ecospat (Di Cola et al., 2017) R packages to implement the niche similarity tests.

For both thresholds, overlapping potential habitats of D. suzukii and L. japonica were identified. The ratio between the land areas of predicted potential distribution of parasitoid and pest in each country was estimated to determine the country-wise biocontrol coverage potential percentage, for both thresholds. Identification of overlapped area and estimation of land area in terms of biocontrol coverage were done in QGIS Tisler desktop version 3.24.3 (QGIS Geographic Information System, 2022). All models were represented in an Eckert III map projection.

Results

For each of the two species, we developed 540 candidate models, of which 510 models for D. suzukii and 533 models for L. japonica were statistically significantly better than random expectations according to the partial ROC tests (P < 0.05). Of the statistically significant models, 53 models for D. suzukii and 11 models for L. japonica were also acceptable in having low (<5%) omission. Finally, based on low model complexity (i.e., low AICc value), our top model for D. suzukii included linear and quadratic feature classes, a relatively low regularization multiplier value (0.6), and four multivariate environmental variables (PC 1–PC 4) (Table 1). Our best model for L. japonica had a higher regularization multiplier value (2.0), and included more multivariate environmental variables (PC 1–PC 7), also with linear and quadratic feature types (Table 1). In the two-species modeling experiment, we developed 540 models, and all models were statistically significantly better than random expectations (P < 0.05). However, none of the models met the omission rate threshold (E = 5%). We found that, even relaxing the threshold (E = 7%) did not result in the selection of any of the two-species models as best model for parasitoid. We therefore confirmed that inclusion of pest model as a predictor variable did not improve model performance for the parasitoid.

Table 1 Model evaluation.

Performance summary of pest, parasitoid, and two-species parasitoid models.

Species	Models	Mean AUC ratio	OR	AICc	
Leptopilina japonica	M_2.0_F_lp_Set_6	1.76	0.00	839.55	
Drosophila suzukii	M_0.6_F_lq_Set_3	1.47	0.04	8365.78	
Two-species	M_1.0_F_l_Set_5	1.79	0.07	798.08	
Notes.

OR-Omission rate. Name of models indicates the details of regularization multiplier value, feature class and environmental dataset.

Collective contributions of PC1 and PC2 to the final model for D. suzukii, and that of PC1, PC3, and PC7 to the final model for L. japonica was >75% (Table 2). For PC1 in the D. suzukii model, no individual climatic variables met the factor loading criterion of 0.35. However, on PC2, variables meeting that criterion included a contrast of precipitation of driest quarter (0.41) and precipitation during driest month (0.40) with mean diurnal range (−0.38). Other variables that fell just short of the threshold were mean temperature of coldest quarter (0.34) and minimum temperature of coldest month (0.34). For the L. japonica model, isothermality (0.82) was the largest contributor to PC7, and PC3 was a contrast of precipitation of wettest quarter (0.40), precipitation during wettest month (0.43), and precipitation seasonality (0.52) with precipitation of driest month (−0.35). Mean diurnal range (−0.34) fell just short of the threshold.

Table 2 Important variables.

Relative contribution of climatic variables to principal components.

Variables	Principal components	
	PC1	PC2	PC3	PC4	PC5	PC6	PC7	
Annual mean temperature	0.33	−0.17	−0.12	0.01	0.15	−0.07	−0.07	
Mean diurnal range	0.10	−0.38	−0.14	0.34	−0.58	0.47	−0.34	
Isothermality	0.32	−0.02	0.00	−0.14	−0.37	0.18	0.82	
Temperature seasonality	−0.32	−0.02	−0.01	0.39	0.15	−0.07	0.29	
Maximum temperature of warmest month	0.25	−0.29	−0.20	0.32	0.27	−0.06	0.10	
Minimum temperature of coldest month	0.34	−0.08	−0.09	−0.17	0.12	−0.06	−0.08	
Temperature annual range	−0.30	−0.11	−0.02	0.48	0.04	0.04	0.19	
Mean temperature of warmest quarter	0.27	−0.26	−0.19	0.25	0.33	−0.16	0.10	
Mean temperature of coldest quarter	0.34	−0.11	−0.08	−0.13	0.06	−0.03	−0.13	
Annual precipitation	0.24	0.34	0.15	0.22	0.03	0.13	0.11	
Precipitation of wettest month	0.25	0.22	0.43	0.24	0.13	0.16	−0.13	
Precipitation of driest month	0.12	0.40	−0.35	0.21	−0.25	−0.31	−0.10	
Precipitation seasonality	0.09	−0.31	0.52	0.11	−0.38	−0.68	−0.02	
Precipitation of wettest quarter	0.25	0.24	0.40	0.24	0.12	0.19	−0.08	
Precipitation of driest quarter	0.13	0.41	−0.33	0.21	−0.22	−0.26	−0.05	
Average contribution of PCs to final model of pest	56.62	36.63	6.49	0.27	–	–	–	
Average contribution of PCs to final model of parasitoid	10.81	2.14	13.42	4.41	6.58	8.07	54.57	
Notes.

Factor loadings ≥ 0.35 are shown in bold.

Our model for D. suzukii predicted potential distributional areas in southern and eastern China, with some extensions towards central Asian regions (Fig. 2). Farther north in Asia, Japan and the Korean Peninsula were predicted to hold broad suitable areas for D. suzukii. Predicted suitable areas covered seven nations [Afghanistan, Pakistan, India, Nepal, China (Tibetan Autonomous Region), Bhutan, and Myanmar] across the entire northwest-southeast spread of the Himalayas. In Oceania, southeastern Australia and much of New Zealand were predicted to hold suitable conditions for D. suzukii invasion.

Figure 2 Ecological niche models.

Predicted distribution of potential distributional areas of Drosophila suzukii and Leptopilina japonica across the world.

Already-invaded parts of western Europe and the southeastern United States were identified as highly suitable for D. suzukii populations, which is logical given that occurrences there were part of the model training data. In South America, the entire geographic extent of Uruguay, known to hold invasive populations, was identified as suitable for D. suzukii; parts of other known-invaded countries (Chile, Argentina, Brazil) were also identified as suitable: eastern and northeastern Argentina, southern Brazil, and western and southern Chile. Peru is the only country in South America predicted to hold suitable areas for D. suzukii invasion for which no invasive populations are known; predicted potential distributional areas spanned the Andean Cordillera.

The modeled potential geographic distribution for L. japonica (Fig. 2) was broad and continuous in Asia, covering southern and northeastern Asian countries (India, China (Tibetan autonomous region), Nepal, Bhutan, North Korea, South Korea, and Japan). Other potential distributional areas were more sparse, in northwestern Europe, western North America, and in western and southern Chile in South America.

Binary models and biocontrol coverage estimation

Binary model outputs were developed for D. suzukii and L. japonica (Fig. 3) to identify presence or absence of the two species in the area of interest. At the 5% threshold, potential presence of D. suzukii was predicted in 162 countries (File S3), covering a total area of ∼4.82 × 107 km2. Potential presence of L. japonica was predicted in 148 countries (File S3), covering a total area of 2.71 × 107 km2. At the 10% threshold, total coverage of predicted area was reduced to 3.44 × 107 km2 in 152 countries for D. suzukii, and 2.46 × 107 km2 in 146 countries for L. japonica (File S3).

Figure 3 Binary models.

Modeled suitable areas for Drosophila suzukii and Leptopilina japonica based on thresholding at E = 5% and E = 10%.

Niches of pest and parasitoid were not demonstrably distinct in either geographic (empirical D = 0.76, P > 0.05) or environmental (empirical D = 0.35, P > 0.05) spaces, as the observed D values fell within the 95% confidence limits of the null distribution of D values in both methods (File S4). As such, no empirical evidence indicates that the two species have distinct ecological niches, and their distributional overlap can be explored as a bellweather of potential for distributional co-occurrence.

Overlapping suitable areas of D. suzukii and L. japonica to identify possible biocontrol regions for both thresholds (Fig. 4) showed that potential for L. japonica- mediated biocontrol management of D. suzukii existed in 125 nations at E = 5%, and 121 nations at E = 10% (Table 3). At a global level, the total possible biocontrol area was estimated to range 1.44 × 107 km2−1.87 × 107 km2 based on the different thresholds. Country-wise biocontrol coverage estimation revealed that about half of the countries (65) had more than 50% biocontrol potential (i.e., area suitable for both fly and wasp; Table 3), with broadest areas in China (∼4.4 × 106 km2), India (∼1.1 × 106 km2), Zambia (4.5 × 105 km2), and Angola (∼4.2 × 105 km2).

Figure 4 Overlapped niches.

Representation of modeled suitable biocontrol areas in terms of overlapping climatic niches of Drosophila suzukii and Leptopilina japonica.

Table 3 Biocontrol coverage.

Modeled potential for biocontrol coverage corresponding to the potential distribution of pest (Drosophila suzukii) and parasitoid (Leptopilina japonica).

Country	Pest distribution (km2)	Overlapping wasp distribution (km2)	Biocontrol coverage (%)	
	E= 5%	E= 10%	E= 5%	E= 10%	E= 5%	E= 10%	
Afghanistan	120756.19	77038.50	87583.33	69587.73	72.53	90.33	
Albania	28019.74	28019.74	27942.43	27810.06	99.72	99.25	
Algeria	444458.49	222597.11	46214.07	43026.86	10.40	19.33	
Andorra	452.25	452.25	407.86	344.49	90.19	76.17	
Angola	725060.74	209636.29	422423.06	152662.75	58.26	72.82	
Argentina	2610531.21	2136682.12	487246.09	326141.97	18.66	15.26	
Armenia	29588.31	27565.71	9573.41	7333.56	32.36	26.60	
Australia	4132533.88	2994970.31	284270.56	200146.28	6.88	6.68	
Austria	83993.20	83993.20	77002.77	73155.41	91.68	87.10	
Azerbaijan	85470.21	82660.14	29145.06	24386.68	34.10	29.50	
Bahamas	9429.07	9429.07	7814.62	7766.95	82.88	82.37	
Bangladesh	128942.02	101548.17	88769.51	75811.84	68.84	74.66	
Belarus	207499.14	207499.14	131398.08	72083.15	63.32	34.74	
Belgium	30597.07	30597.07	27115.20	24893.18	88.62	81.36	
Bhutan	38954.11	37112.34	33859.02	32179.91	86.92	86.71	
Bolivia	475959.05	203932.55	3394.30	1816.23	0.71	0.89	
Bosnia and Herzegovina	51824.53	51824.53	32450.62	28581.47	62.62	55.15	
Brazil	2088214.01	1385571.53	365390.65	278811.34	17.50	20.12	
Brazilian Island	2.82	2.82	2.82	2.82	100.00	100.00	
Bulgaria	112513.51	112513.51	1544.03	971.72	1.37	0.86	
Cabo Verde	1750.55	630.75	479.08	269.84	27.37	42.78	
Cambodia	6253.09	790.65	2055.16	133.61	32.87	16.90	
Cameroon	26383.63	1809.00	14635.67	169.68	55.47	9.38	
Canada	2793734.16	2155273.21	565894.51	487960.16	20.26	22.64	
Chile	582097.31	475909.40	205933.41	193145.19	35.38	40.58	
China	4488161.45	3862635.87	4374430.56	3827973.12	97.47	99.10	
Colombia	91947.38	68214.15	340.13	340.13	0.37	0.50	
Croatia	52932.84	52932.84	40105.39	36135.71	75.77	68.27	
Cuba	81360.79	7022.89	510.20	26.30	0.63	0.37	
Cyprus	5122.47	3433.88	4427.71	3433.88	86.44	100.00	
Cyprus No Mans	296.73	33.20	72.49	33.19	24.43	99.98	
Czechia	78758.87	78758.87	68665.32	58491.87	87.18	74.27	
Democratic Republic of the Congo	420374.68	150483.68	161159.34	75871.68	38.34	50.42	
Denmark	202079.28	130500.31	166506.11	111597.35	82.40	85.52	
Djibouti	13137.52	–	467.25	–	3.56	–	
Egypt	180965.62	–	120.06	–	0.07	–	
Equatorial Guinea	14.17	14.17	14.17	14.17	100.00	100.00	
Estonia	44389.34	44389.34	43945.97	37804.22	99.00	85.17	
Ethiopia	413796.88	155614.76	39007.25	21656.50	9.43	13.92	
Finland	300806.37	238124.40	93531.28	50354.32	31.09	21.15	
France	562246.74	556671.13	274309.40	231951.91	48.79	41.67	
Gabon	45201.71	2552.27	1222.11	267.10	2.70	10.47	
Georgia	69301.13	69301.13	61604.07	58319.71	88.89	84.15	
Germany	355684.24	355684.24	192739.07	159474.97	54.19	44.84	
Greece	123576.16	123576.16	84346.67	79969.84	68.25	64.71	
Guatemala	22913.49	16722.41	2064.41	1734.17	9.01	10.37	
Guinea	4192.93	83.65	3857.20	83.65	91.99	100.00	
Hungary	93200.95	93200.95	7046.93	3609.03	7.56	3.87	
Iceland	98272.76	94978.48	88865.07	83482.50	90.43	87.90	
India	1274811.80	685789.21	1102039.55	543419.47	86.45	79.24	
Iran	336850.53	116130.87	63329.32	37732.16	18.80	32.49	
Iraq	34709.55	21099.93	29371.04	19865.20	84.62	94.15	
Ireland	66629.91	66332.38	55507.01	52328.26	83.31	78.89	
Israel	13449.48	8745.65	11881.97	8745.68	88.35	100.00	
Italy	295635.80	295613.28	264427.13	250415.73	89.44	84.71	
Japan	357893.96	356945.12	357893.91	356945.07	100.00	100.00	
Jordan	4921.88	507.56	1872.74	507.56	38.05	100.00	
Kazakhstan	201132.14	41577.90	706.08	190.06	0.35	0.46	
Kosovo	10913.08	10913.08	1765.30	1373.59	16.18	12.59	
Kyrgyzstan	90256.07	43305.48	1333.13	63.53	1.48	0.15	
Laos	186149.10	124451.87	149422.53	103577.26	80.27	83.23	
Latvia	64162.08	64162.08	60919.43	54215.14	94.95	84.50	
Lebanon	9800.04	8682.25	9001.50	8503.74	91.85	97.94	
Lesotho	30106.52	30106.52	318.96	28.00	1.06	0.09	
Libya	184279.76	37328.30	5699.73	4033.72	3.09	10.81	
Liechtenstein	137.25	137.25	137.25	137.25	100.00	100.00	
Lithuania	64816.37	64816.37	40725.99	27301.77	62.83	42.12	
Luxembourg	2608.47	2608.47	2608.47	2328.89	100.00	89.28	
Madagascar	187519.17	104765.96	79329.34	39438.11	42.30	37.64	
Malawi	111209.70	74340.02	108138.07	72489.54	97.24	97.51	
Malta	270.90	270.90	270.90	270.90	100.00	100.00	
Mauritius	1802.79	94.64	1752.24	44.08	97.20	46.58	
Mexico	691072.40	296644.27	251952.23	125678.99	36.46	42.37	
Moldova	33206.48	33206.48	4811.96	1963.38	14.49	5.91	
Monaco	3.96	3.96	3.97	3.97	100.00	100.00	
Montenegro	13631.45	13631.45	11836.29	11080.44	86.83	81.29	
Morocco	354222.81	165793.77	27266.40	23178.86	7.70	13.98	
Mozambique	278561.34	73923.95	154695.03	54111.84	55.53	73.20	
Myanmar	445821.06	375179.90	361982.68	301972.44	81.19	80.49	
Namibia	66288.68	–	483.94	–	0.73	–	
Nepal	145624.62	141915.07	121958.80	119500.60	83.75	84.21	
Netherlands	36761.20	36260.59	17922.53	15136.66	48.75	41.74	
New Zealand	217910.77	212122.44	96771.30	83214.96	44.41	39.23	
Nigeria	15934.24	566.67	12716.29	337.88	79.80	59.63	
North Korea	120894.87	118933.68	120894.94	118933.71	100.00	100.00	
North Macedonia	25385.27	25385.27	3664.82	3358.17	14.44	13.23	
Northern Cyprus	2290.99	429.38	1414.83	429.39	61.76	100.00	
Norway	285817.32	263207.96	225627.24	206147.09	78.94	78.32	
Oman	28552.55	1859.00	3159.53	1543.99	11.07	83.06	
Pakistan	153944.54	83124.10	85069.07	49372.57	55.26	59.40	
Paraguay	341582.31	230965.19	37691.58	29169.69	11.03	12.63	
Peru	495995.98	431000.71	49307.80	45336.48	9.94	10.52	
Poland	312841.88	312841.88	214155.94	156484.69	68.46	50.02	
Portugal	89463.23	89463.23	73702.57	68237.60	82.38	76.27	
Republic of Serbia	77628.71	77628.71	8713.93	5548.68	11.23	7.15	
Republic of the Congo	72181.30	6283.59	446.79	26.96	0.62	0.43	
Romania	235895.13	235895.13	79961.95	62526.40	33.90	26.51	
Russia	3710679.26	2033909.21	662712.71	395046.93	17.86	19.42	
San Marino	60.32	60.32	60.32	60.32	100.00	100.00	
Slovakia	48457.79	48457.79	25232.78	22596.49	52.07	46.63	
Slovenia	20295.63	20295.63	19931.57	19671.10	98.21	96.92	
South Africa	902797.42	591473.81	7170.51	5131.33	0.79	0.87	
South Korea	94652.90	94652.90	94652.97	94652.97	100.00	100.00	
Spain	502618.58	495648.00	136222.35	112053.05	27.10	22.61	
Sudan	2656.20	–	83.33	–	3.14	–	
Sweden	436539.07	378204.51	158054.75	115334.67	36.21	30.50	
Switzerland	41344.82	40890.87	37361.01	36701.13	90.36	89.75	
Syria	22207.60	12452.65	14768.48	12044.15	66.50	96.72	
Taiwan	24849.72	20605.88	21861.78	18151.55	87.98	88.09	
Tajikistan	73944.07	45547.87	38519.69	25750.20	52.09	56.53	
Thailand	117459.90	22050.79	103622.02	22024.89	88.22	99.88	
Tunisia	78947.05	53298.18	10701.95	8871.64	13.56	16.65	
Turkey	761388.06	678724.38	257025.95	224311.08	33.76	33.05	
Turkmenistan	59106.76	4758.25	160.83	160.83	0.27	3.38	
Ukraine	570440.95	570440.95	182074.04	152631.46	31.92	26.76	
United Arab Emirates	428.91	15.84	155.89	15.84	36.35	100.00	
United Kingdom	247439.54	243271.25	134076.15	124951.47	54.19	51.36	
Tanzania	540638.96	197481.86	110629.43	79486.65	20.46	40.25	
United States of America	7267495.86	5752263.93	2879003.63	2546609.52	39.61	44.27	
Uruguay	176465.55	176465.55	75249.12	56325.26	42.64	31.92	
Uzbekistan	64090.46	22578.88	12752.88	9003.24	19.90	39.87	
Vatican	0.01	0.01	0.01	0.01	100.00	100.00	
Vietnam	208079.26	173719.30	179340.06	158519.90	86.19	91.25	
Zambia	471456.61	148179.91	452563.86	146359.35	95.99	98.77	
Zimbabwe	186694.49	78523.17	135058.40	69964.04	72.34	89.10	
Notes.

E indicates thresholding level.

Discussion

Extremely fast range expansion as a consequence of globalization (Iacovone et al., 2015), with severe economic damage to the fruit trade industry (Bolda, Goodhue & Zalom, 2010; Gabarra et al., 2015), has led to efforts to model ecological niches and predict potential distributions for D. suzukii both locally (Castro-Sosa et al., 2017; De la Vega & Corley, 2019) and globally (Santos et al., 2017; Ørsted & Ørsted, 2019; Reyes & Lira-Noriega, 2020). Comparing with previous global-scale models, our models predicted highly suitable areas for D. suzukii most similar to the model developed by Ørsted & Ørsted (2019), and less similar to those of Santos et al. (2017) and Reyes & Lira-Noriega (2020). Relatively broad geographic areas in the southern part of central and eastern Africa were predicted to be suitable in the models developed by Santos et al. (2017) and Reyes & Lira-Noriega (2020) compared to our model and that of Ørsted & Ørsted (2019). Unlike the predictions of Santos et al. (2017) and Reyes & Lira-Noriega (2020), Patagonian region of Argentina was not included as suitable habitat for D. suzukii in our model and that of Ørsted & Ørsted (2019). Another major difference between our model and those of Santos et al. (2017) and Reyes & Lira-Noriega (2020) is that their models predicted a large extent of eastern India as suitable habitats for D. suzukii. However, according to our model, the suitability was more prominent in far north, and also in some parts of Western Ghats in southern India. Although similar in many aspects of predicted distributions, our model differed notably from that of Ørsted & Ørsted (2019) in predicting the east–west continuity of potential distribution of D. suzukii in United States as our model showed a discontinuous distribution of potential habitats.

In exploring bioclimatic variable contributions to the pest model, mean temperature of coldest quarter and minimum temperature of coldest month both had contributions to the models (Table 2) that were substantive enough to merit comment; similar observations of the influence of cold temperatures on D. suzukii distribution were made by Ørsted & Ørsted (2019). Limiting influence of winter temperatures on the establishment of D. suzukii populations is evident from the facts that prolonged low temperature exposure (<10°) is detrimental for its viability (Dalton et al., 2011). A recent meta-analysis (Ørsted et al., 2021) revealed that temperature extremes are highly significant in determining the survival and population activity of the species. Preference of D. suzukii for humid environments (Gutierrez, Ponti & Dalton, 2016; Santos et al., 2017; Ørsted & Ørsted, 2019) was reflected in our models via high contributions of precipitation of driest quarter and precipitation of driest month. Large contributions of isothermality, precipitation seasonality, precipitation of wettest month, precipitation of driest month, and precipitation of wettest quarter to L. japonica model (Table 2) indicate that temperature fluctuations and humidity of environments may also play crucial roles in constraining the distribution of L. japonica.

For obvious reasons, choosing biological control agents for D. suzukii that have niche preferences similar to those of the fly will be helpful (Robertson, Kriticos & Zachariades, 2008; Olfert et al., 2016; Tepa-Yotto et al., 2021a; Tepa-Yotto et al., 2021b) in the global-scale biological control challenge. Matching the climatic niche requirements of pest and parasitoid will increase chances of long-term establishment of the parasitoid across key regions (Robertson, Kriticos & Zachariades, 2008), resulting in more successful management via biological control. Despite various previous studies modeling the climatic niche of D. suzukii, to the best our knowledge, no effort has been made so far to study the potential distribution of climatic niches of any parasitoid of D. suzukii in combination with analyses of the climatic niche of the fly pest.

Range expansion of D. suzukii in Europe and North America occurred after initial outbreaks in California, Spain, and Italy, all in 2008 (Rota-Stabelli, Blaxter & Anfora, 2013; Asplen et al., 2015). Niche filling related to absence of competitors or natural enemies, high adaptability to temperate climates, high dispersal ability, and high reproductive output, are major factors contributing to the unprecedented invasion of D. suzukii (Rota-Stabelli, Blaxter & Anfora, 2013). As niche filling is an important factor, assessing the geographic distribution of climatic niches of D. suzukii becomes an indispensable step in biological control programs, as it can provide an initial estimate of the geographic limits for successful parasitoid release (Puppato et al., 2020). Development of niche models for parasitoids, and identification of geographic regions exhibiting overlapping climatic niches between pest and parasitoid, further delimits regions for parasitoid release, making field trials involving elaborate and time-consuming experiments more economical (Sun et al., 2017).

In its native distributional areas, L. japonica is one of most abundant potential parasitoids of D. suzukii (Kimura & Novković, 2015; Puppato et al., 2020); its occurrence in Europe(Puppato et al., 2020) and North America (Abram et al., 2020; Abram et al., 2022; Beers et al., 2022) was identified only recently. Previous laboratory experiments in the United States indicated that South Korean L. japonica strains attacked the North American strains of D. suzukii readily (Daane et al., 2016), supporting at least in part the suitability of L. japonica as a biocontrol agent for D. suzukii. Although occurrence records of L. japonica were scarce, our modeled climatic niche for L. japonica overlapped broadly with that of D. suzukii in known-invaded regions (Figs. 2–4), meeting one of the major ecological requirements for a ‘natural enemy species’ to be a candidate biological control agent (Robertson, Kriticos & Zachariades, 2008; Olfert et al., 2016).

In addition, our statistical quantification of similarity or difference in ecological requirements of pest and parasitoid failed to reject the null hypothesis of niche similarity (Peterson, 2011; Tocchio et al., 2015), revealing that their overlapping niches are similar, at least given the data available to us. Using overlap of potential distributions in geographic space, Tepa-Yotto et al. (2021b) explored the possibility of ENM in devising biological control measures for the fall armyworm, Spodoptera frugiperda, as regards its key parasitoids. In terms of the Hutchinsonian duality, overlapping potential distributions of pest and parasitoid in geographic space alone does not help researchers to conclude that ecological requirements of the species are same because a point in geographic space can be expressed as only one point in environmental space but a point in environmental space may expressed by more than one point in geographic space (Castaneda-Guzman, 2022; Nuñez Penichet et al., 2022). Considering niche overlap of pest and parasitoid in both geographic space and environmental space is therefore essential to confirm that the species can indeed interact. Ability of the biocontrol agent to colonize the full distributional area of the target species is critical for the success of biocontrol programs (Gupta et al., 2022). The two-species model developed for gaining insight into the biotic interactions shaping the potential geographic distribution of L. japonica underperformed compared to the climate-only model. These results thus contrasted with previous findings highlighting the importance of including biotic predictors in ecological niche modeling procedures to improve model performance (Araújo & Luoto, 2007; Giannini et al., 2013; Dormann et al., 2018; Simões & Peterson, 2018; Bebber & Gurr, 2019; Ashraf, Chaudhry & Peterson, 2021).

We recommend a niche-based, target-oriented prioritization approach in designing biological control programs aimed at D. suzukii. In Europe, three interlinked factors, (1) recently recorded occurrences (Puppato et al., 2020), (2) predicted suitability in 17 European countries (∼39% of European countries) with biocontrol coverage of >80% at both thresholds (E = 5% and E = 10%)(Table 3), and (3) increasing consumer preference towards organic fruits (Murphy et al., 2022), make L. japonica a candidate parasitoid for control of D. suzukii. In the remaining European countries, in particular those exhibiting biocontrol coverage <50%, we suggest extra care in defining appropriate geographic boundaries for L. japonica release plans (Table 3). In the United States and Canada, the potential distribution of L. japonica overlapped only one-third of D. suzukii’s potential distributional area, demanding strict site-specific release planning. Site-specific pest management utilizing pest distributional information is preferred over uniform pest management (Park, Krell & Carroll, 2007). However, for effective site-specific biological control of pests, not only the pest distributional information but also the niche overlap between pest and parasitoids must be taken into account. Irrespective of the biocontrol coverage in D. suzukii invaded regions, any L. japonica release strategy has to rely not only on specific details of both site and niche considerations but on host-specificity trials by taking non-target insects from the local fauna also into account (Van Driesche & Hoddle, 1997).

In a study that tested the host specificity of three parasitoids of D. suzukii (Daane et al., 2021), highly host-specific G. brasiliensis (Wang et al., 2020; Daane et al., 2021) successfully parasitized the target D. suzukii and three other related species (D. simulans, D. melanogaster, and D. persimilis). Our species of interest (L. japonica) was successful in parasitizing D. suzukii at a high rate, as well as nine other related species (D. simulans, D. melanogaster, D. persimilis, D. montana, D. robusta, D. tripunctata, D. willistoni, D. funebris, Hirtodrosophila duncani) at relatively lower rates. Taking these facts into consideration, prior to development of L. japonica release plans, care must be taken to properly address two important questions: what is expected range of hosts to be parasitized, and what are the ecological and economic values that we place on them (Van Driesche & Hoddle, 1997), as well as real-world performance of the parasitoid in small-scale release experiments.

Gross national income (GNI) per capita of the exporting country and incidences of interception of insect pests at international ports of entry are known to be negatively correlated (Liebhold et al., 2006), such that lower-income countries are at a greater risk of pest-induced crop damage (Gaiha et al., 2009) due to poor surveillance (Liebhold et al., 2006). More than 80% of current low-income economies are in Africa (https://datahelpdesk.worldbank.org, accessed on 21/01/2023); based on the biocontrol coverage explored in this paper, the possibility of applying L. japonica mediated biocontrol measures exists in six low-income African countries (Republic of the Congo, Guinea, Madagascar, Malawi, Mozambique, Zambia). Considering the low GNI per capita, and significant export potential of fruits from Malawi (tropical fruits; US$1.41M/year) and Mozambique (tropical fruits; US$3.77M/year) (The observatory of economic complexity data, https://oec.world, accessed on 21/01/2023), D. suzukii invasion in these sub-Saharan African (SSA) countries can affect not only the fruit yield within these countries but also fruit cultivation of the countries of import. Biocontrol measures reduce insect pest multiplication and benefit crop yields in SSA, and large-scale biocontrol programs can enhance food security in this region (Ratto et al., 2022). We recommended extensive field surveys in Malawi and Mozambique to check for presence of D. suzukii; if presence is confirmed, adopting biocontrol strategies in the national and regional farming policies of these countries may produce the double benefit of increasing crop yield within the countries, and reducing possible transnational crossing of D. suzukii to the countries of import. Biological control in SSA comes under the purview of the consultative group on international agricultural research (CGIAR) (Adenle, Wedig & Azadi, 2019). Given that the presence of D. suzukii has already been confirmed in SSA (Kwadha et al., 2021) and that domestic research facilities are relatively poor (Pal, 2011), intervention of multilateral development agencies like CGIAR in detecting invaded insect pests in agricultural fields, estimating its abundance, and devising effective strategies of biological control is recommended (Adenle, Wedig & Azadi, 2019).

In conclusion, this study illustrates a cost-effective pre-assessment strategy that can be applied to any biological control management program before beginning the labor-intensive, time-consuming, and expensive field experiments. Availability of a greater number of occurrence records of L. japonica would further enhance the understanding of the distributional potential of this potential biocontrol agent worldwide. Of course, we do not recommend straight-away release of L. japonica into the fields where biocontrol of D. suzukii may be potentially beneficial to the farming community. We suggest instead to treat this study as a preliminary platform in general to develop a niche-based, target-oriented prioritization approach to select potential species for biocontrol management with the support of evidences from host range trials involving choice and no-choice considerations.

Supplemental Information

File S1 Occurrence records of invasive pest D. suzukii

Presence points of D. suzukii used for global-scale ecological niche modeling

Click here for additional data file.

File S2 Occurrence records of parasitoid L. japonica

Presence points of parasitoid L. japonica used for global-scale ecological niche modeling

Click here for additional data file.

File S3 Estimation of country-wise area of potential habitats

Country-wise area estimation of potential habitats for both pest and parasitoid at most desirable (E = 5%) and maximum permitted (E = 10%) thresholds.

Click here for additional data file.

File S4 Niche similarity testing

Niches of pest and parasitoid are represented along the first two principal components (top panel). Dark shading indicates the occurrence densities of pest and parasitoid by cell. The solid and dashed contour lines indicate 100% and 50% of background environment in the area M. The empirical D value fell within the null distribution of D values generated in both tests (Broennimann et al., 2012, bottom left; Warren, Glor & Turelli, 2008, bottom right), indicating the non-rejection of the null hypothesis of niche similarity. E-space: Environmental space, G-space: Geographic space

Click here for additional data file.

The authors thank the KUENM group for helpful discussions, and the Biodiversity Institute, University of Kansas, for providing facilities for this work. We also thank Marlon E. Cobos, Claudia Nuñez Penichet, Fernando Machado-Stredel, and Jacob C. Cooper for valuable suggestions in implementing the methodology.

Additional Information and Declarations

Competing Interests

Author Contributions

Data Availability

The authors declare there are no competing interests.

Rahul R. Nair conceived and designed the experiments, performed the experiments, analyzed the data, prepared figures and/or tables, authored or reviewed drafts of the article, and approved the final draft.

A. Townsend Peterson conceived and designed the experiments, analyzed the data, prepared figures and/or tables, authored or reviewed drafts of the article, and approved the final draft.

The following information was supplied regarding data availability:

The presence points used for modeling potential habitats of pest and parasitoid, and the country-wise area calculations associated with different thresholds (5% and 10%) for both pest and parasitoid are available in the Supplementary Files.

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
