# Peer review of "Mapping the global distribution of invasive pest Drosophila suzukii and parasitoid Leptopilina japonica: implications for biological control"

_PeerJ, doi:10.7717/peerj.15222_

## Round 0.1 · original submission · Major Revisions

Dear Dr. Nair,

After this first review round, your manuscript was considered to be publishable in PeerJ after major reviews are considered. Please take special attention to the issues raised by both reviewers, but specially Reviewer #1, who raised important questions about the "current knowledge, the results obtained and the conclusions of the authors."

Please prepare a new version of the manuscript along with a rebuttal letter informing about the performed changes and justifying those that were kept unchanged in the new version of the manuscript. I am sure this will significantly help the reviewers to become more confident (or not) about the presented results.

Sincerely,
Daniel Silva

Reviewer 1 ·

Basic reporting

Dear authors and editor,

The article is a study about the overlapping of two species, one pest and another, a parasitoid that is intended to be used as a biological control. The article presents the overlapping of two potential distributions of two species, a pest species and a parasitoid species that is intended to be used as a biological control. This is done using species distribution models.

The article mostly meets the editorial criteria. I think the figures should be adapted to a better projection for a better understanding and presentation of them.

I have reviewed the article in depth and considered both the species ecological niche modelling approach and the pest and biological control approach.

Under the first approach, I consider that not all the necessary information has been extracted from the models. Ecological analyses can be done that would help to better understand and generate knowledge of this. Right now there are only two overlapping distributions and the percentage of overlap, but there is no information about how much the two species' niches are similar, for example. This ecological part is very relevant for the second approach to pests and biological control. The authors could easily address this part, further enriching their contribution.

Under the other approach, that of pests and biological control, I think the statements are very risky and based on not very solid evidence. Claiming that the species as biological control is promising based solely on overlapping distributions is risky. The critical point is that the article shows that the parasitoid is not specific to the pest species but rather is a generalist (The authors verify that the distribution of the target species (resource) is not a variable to consider in the parasitoid model). Releasing a new species to the environment that has a wide host range can negatively impacts on native biological communities. The authors overlook this and the article as it stands now seems to support the release of this species. It is commented that it is a previous article and that more tests should be done from this, but it should appear explicitly commented that without host range experiments and a perspective closer to the conservation of native fauna, no decision can be supported. of liberation. It is not the first case that host range trials are not carried out and biological control species are released because there are articles that support a candidate. The authors should reshape the discussion and be very careful with their contributions on this topic.

Experimental design

I don't have too many drawbacks in this part, the article follows the criteria points. The methodology is very good. The parasitoid species has few records, which although it may worsen the results, it is not possible to obtain more records. Records of parasitoid species are generally scarce and difficult to obtain due to their inconspicuous way of life. It is something that the authors already consider and it seems correct to me. The only debatable thing is the use of variable factors as predictor variables. It is possible that the "pure effect" of the variable is lost by creating factors that are often difficult to interpret. Perhaps a more direct approach with bioclimatic variables can help to understand and generate knowledge of the ecology of the species and is easier to interpret. I think that in this way the ecological information could be more easily analyzed to further enrich the article.

Validity of the findings

This is where I see the most problems with the article. There are differences between current knowledge, the results obtained and the conclusions of the authors.
The previous bibliography mentions a more specific species of the pest, but it is wanted to use L. japonica because it is more common and is more productive, the results indicate that there does not seem to be a direct relationship between the parasitoid and the pest, indicating that it has a wide host range, that the models do not have high suitability in areas where the pest is (see the case that I highlight in the review) and in the conclusions they mention that it is a promising species as a biological control. It seems that they have only taken into account that there is an overlap in the territories. Is this enough to support that L. japonica is a good biological control of D. suzukii? I repeat that, without evidence of the host range of this parasitoid (many times the species that most parasitizes a target species is used, but it is not verified if it is the most effective or the most abundant in the biological community/more generalist) , the statements must be softened a lot and take into account the perspective that the introduction of a new exotic species is being proposed.
The authors should reformulate the discussion considering these perspectives.

Additional comments

Despite the fact that the article has great potential and I liked it, I find that it has deep conceptual problems and the discussion should be further worked on. In addition to the minor revisions that I provide in my review, the authors should focus on two main points: 1) make an ecological analysis comparing the niche of both species and 2) reformulate and broaden the discussion with the perspectives that I have commented. especially lowering the level of the statements referred to biological control.

Annotated reviews are not available for download in order to protect the identity of reviewers who chose to remain anonymous.

·

Basic reporting

First up, sorry for my delay, this is my responsibility, not the Editorial team.
Nair and Peterson perform environmental niche modelling for the invasive pest, Drosophila suzukii and a parasitoid Leptopilina japonica to investigate the global potential for biocontrol in the overlapping geographical range representing the fundamental niches of the two species. I think the introduction is well-structured, clearly written and shows a broad knowledge on existing literature both on the pest and parasitoid. The discussion is well written and provides adequate comparison with previous studies. Overall, a nice addition to pest management literature.

Experimental design

The last paragraph clearly states the chronological objectives of the paper. The ENM methods are state-of-the-art (although I have one objection, see below). I really like the candidate models and filtering of models.

Validity of the findings

The ENM model results of D. suzukii do not bring much new information given a suite of recent similar models, but the overlapping country-wise biocontrol coverage is interesting, although results are only presented as predicted binary ranges without regard to possibilities for actual biocontrol in each or a few select countries. While I agree that the authors’ approach can be cost-effective pre-assessment strategy, I suggest discussing differences in possibilities for actually implementing such a biocontrol program due to country differences in economic status, crop density, organic vs conventional production etc. Otherwise, I think the results are nicely presented. Conclusions are well stated and linked to the last paragraph of the introduction.
I have issue with the occurrence data provided (Suppl. tables 1 and 2), they contiain no references. I know they are obtained from a selection of online databases, but they should provide references for individual occurrence points.

Additional comments

My only real issue is with transforming climatic variables into PC multivariate environmental variables. I appreciate that the authors do this to minimize collinearity (and dimensionality, but why is this important?). A large part of the utilization of MaxEnt and other ENM algorithms is interpretation of response curves and ranking of variable importance, however, PCA makes such interpretation of the climatic variables impossible. While it may be more methodologically sound (although I am not convinced), I think the ecological relevance/interpretation is lost, which to me is a bad trade-off. At the same time, there are plenty other regularly used approaches for reducing collinearity between explanatory variables. Therefore, I think the authors should re-consider this approach, or at the very least discuss the implications of reducing climatic variables to PCs, and how it limits the interpretation of the model output. Some of same niche modelling papers that the authors already refer to use the model output to investigate which climatic factors are most important in limiting the distribution (and population abundances, see e.g. Ørsted et al. 2021 Pest Manag Sci 77:4555-4563. doi: 10.1002/ps.6494), which could be useful to include, given the lack of interpretability of the multivariate climatic variables.

All the best,
Michael Ørsted

Minor details:
Figure 1: I do not have access to the high-res figures, but it can be hard to distinguish L. japonica occurrences as red dots against a orange background (and red dots behind blue, maybe reverse?). I suggest different colors that are more accommodating for e.g. color-blind people.
Figure 2: I know suitability is probably >0 everywhere, but I suggest setting a lower threshold (0.01 or similar) and give those areas a neutral color like grey or something, it can be hard to distinguish important suitability areas given that the color gradient is omnipresent.

---

## Round 0.2 · Minor Revisions

Dear Dr. Nair,

thank you for your patience and congratulations regarding the improvements to your manuscript. After this new review round, Reviewer #2 still believes some improvements could be made to a new version of your manuscript. I agree that some of the suggested additions are beneficial to the manuscript. Therefore, I believe they should be included in it.

Specifically, I often also use PCA in my manuscripts, and I believe you can keep this analysis in the manuscript. Still, some additional discussion about its use, as indicated by the reviewer, may be welcome.

Please provide a new version of your manuscript whenever it is possible, along with a rebuttal letter explaining the included changes. As soon as Reviewer #2 is satisfied with the few improvements to be made or with the provided explanations for not implementing the changes in the new version of your manuscript, I will also be satisfied.

Sincerely.
Daniel Silva

·

Basic reporting

In the revised version, the authors have sufficiently addressed the concerns I had in the original version. I particularly liked the implementation of the niche similarity analysis (as suggested by reviewer #1), although I think the authors could have elaborated on the results, particularly in the discussion, have others found similar results (perhaps for other host/parasitoid species)? what are the strengths and weaknesses of such an analysis, what does it really show? I also liked the addition of assessing the risk of invasion and economic impact in Africa, as well as the revised figures.
With a few tweaks, I think the revised version could be made ready for publication.

Experimental design

I have a few remaining comments:
- I think we still disagree on transforming climatic variables into PC multivariate environmental variables (as also highlighted by reviewer #1), and that “determining important variables for a species is subjective. We adopted PCA to reduce the dimensionality and multi-collinearity, and thus avoid model over-fitting”. There are still other numerous ways of reducing multi-collinearity, one of which is to simply remove correlated variables prior to ENM analysis. I appreciate the authors using factor loadings as proxies for variable importance, although the arbitrary threshold can be discussed, and I don’t think factor loadings are any easier to interpret. For instance, the authors use contrasting loadings on individual PC axes, but how do PCs compare between species? And what does it mean that loading for precipitation during driest month is +0.40 for D. suzukii and -0.35 for the L. japonica, is it even comparable? This could be discussed in greater detail in my opinion.

- I understand the results of the niche-similarity analysis, but I think the phrasing is wordy and tricky and could be made simpler, e.g. in the discussion: “In addition, statistical quantification of the similarity in the ecological requirements of pest and parasitoid has led to non-rejection of the null hypothesis of niche similarity (Peterson, 2011; Tocchio et al., 2015), revealing that their overlapping niches are not demonstrably distinct…” (Lines 415-417). I realise that you’re not actually verifying a hypothesis, just failing to reject the null, but this is placed in the discussion, and I think it would be okay here to simplify such a statement for the reader.

Validity of the findings

No additional comments (see above)

Additional comments

3) I had a minor comment for the first version, which unfortunately did not make it in my review, my apologies: Table 1: Maybe I missed it, but I cannot find a description of how Mean AUC ratio in Table 1 is calculated and info on how it should be interpreted, preferably with a reference.


All the best,
Michael Ørsted

---

## Round 0.3 · accepted · Accept

Dear Dr. Nair,

After this new review round, I personally checked the responses and improvements made to the manuscript. I believe you successfully addressed all the issues the reviewer raised. So, I did not submit the manuscript to a new read by the reviewer.

I am happy with the final version of the manuscript as it is. Therefore, it is a pleasure to inform you that your manuscript has formally been accepted for publication in PeerJ! Congratulations!

Daniel Silva